# The Singapore Physical Activity Readiness Questionnaire 2021 (SPARQ 2021)—Results of Public Feedback

**DOI:** 10.3390/healthcare13151837

**Published:** 2025-07-28

**Authors:** Tess Lin Teo, Ian Zhirui Hong, Lisa Cuiying Ho, Stefanie Hwee Chee Ang, Anantharaman Venkataraman

**Affiliations:** 1Department of Emergency Medicine, Singapore General Hospital, Singapore 169608, Singapore; tess.teo.lin@singhealth.com.sg (T.L.T.); ian.hong.zr@gmail.com (I.Z.H.); 2Department of Orthopaedic Surgery, Sengkang General Hospital, Singapore 544886, Singapore; ho.lisa.ho@gmail.com; 3Sport Singapore, Singapore 397630, Singapore; stefanie_ang@sport.gov.sg

**Keywords:** self-administered pre-participation survey, screening, physical activity, SPARQ, co-morbidities, difficulty in understanding

## Abstract

*Introduction*. Singapore had previously embraced at least two types of pre-participation questionnaires for those intending to take up or enhance their level of physical activity (PA). Concern over the usefulness of and difficulty in understanding these questions led to the design of a Singapore Physical Activity Readiness Questionnaire (SPARQ). The primary objective of this study was to review the level of difficulty in understanding the seven SPARQ questions. Secondary objectives included the rate of identifying individuals as unfit for PA and to seek public feedback on this tool. *Method*. A public, cross-sectional survey on the SPARQ was carried out, obtaining participants’ bio-characteristics, having them completing the SPARQ and then providing feedback on the individual questions. *Results*. Of the 1136 who completed the survey, 35.7% would have required referral to a medical practitioner for further evaluation before the intended PA. Significant difficulty was experienced with one question, moderate difficulty with four and only slight difficulty with the remaining two. The length of the questions and use of technical terms were matters of concern. Suggestions were provided by the participants on possible amendments to the questions. *Conclusions*. The very high acceptance rate of the SPARQ will need to be tempered with modifications to the questions to enhance ease of understanding and use by members of the public.

## 1. Introduction

Sports injuries can occur during competitive or recreational physical activity (PA), usually through falls or cardiovascular collapse. Regular PA also has many benefits, such as lower rates of obesity, heart disease and a better quality of life. Adults are usually encouraged to be physically active on most days of the week to better achieve the expected health benefits of regular exercise [1]. The United Kingdom Chief Medical Officers’ physical activity guidelines and the American College of Sports Medicine (ACSM) recommend 150 min of moderate-intensity aerobic exercise and 2 or more days per week of strengthening activity for adults and the elderly and 60 min of moderate/vigorous activity a day for children [2,3]. For Asian populations, modifications of such PA requirements have been proposed [4]. The World Health Organization (WHO) recommends at least 150–300 min of moderate-intensity PA weekly [5].

To promote early identification of persons prone to injuries, pre-participation screening systems have been suggested for use by those wishing to participate in moderate or intense exercise or increase their level of PA [6,7]. These are intended to identify individuals with a likely risk for sudden cardiac death (SCD) or other major injury. Such at-risk individuals should require medical evaluation before being cleared for a PA program. One such system, when applied to adults over 40 years old, resulted in nearly 95 percent of those in that age group being referred to a physician before engaging in any form of exercise [8,9]. Refinement of preparticipation screening procedures in 2015 resulted in an approximately 41% decrease in the proportion of such referrals [10].

In Singapore, over the years, different pre-participation questionnaires have been suggested by the Sports Safety Committee, beginning in 2007 with the Physical Activity Readiness Questionnaire (PAR-Q) followed by the Get Active Questionnaire (GAQ) in 2019 [6,7,11,12,13]. However, concern has been expressed about the complexity and difficulty in self-administering these tools and that they did not address the occurrence of heat-related injuries for those exercising in hot and humid environments [14,15,16].

To address these concerns, the authors drafted a fresh pre-participation questionnaire for members of the public. Provisionally named as the Singapore Physical Activity Readiness Questionnaire 2021 (SPARQ—Figure 1), this document had seven questions, each needing either a Yes or No answer. A Yes answer to any of the seven questions would suggest the participant is unfit for PA and require further medical evaluation. The readability test scores for this questionnaire were Flesch–Kincade Readability Grade Level = 9 and Flesch Readability Ease score 45.2.

The first question was “Do you have either high blood pressure or a heart condition for which you still require treatment and close follow-up by a doctor?”. This was intended to address those whose high blood pressure or heart conditions were not well controlled, potentially increasing the risk of cardiac arrest if they were to increase their exercise intensity.

The second question, “Do you have moderate or severe joint pains made worse by physical exercise?”, was to identify those with disabling symptoms that were likely aggravated by increasing PA or made one prone to injury.

The third question, “Have you been feeling unwell over the last one week with either fever, sore throat, cough, vomiting or diarrhoea?”, was to address those acutely unwell and more prone to heat-related injuries.

The fourth question, “Do you get chest pains either during or after physical activity?”, was to identify those with anginal symptoms requiring further evaluation and stabilization before being advised on PA. This was to facilitate early identification of sudden cardiac arrest-prone individuals.

The fifth question, “Do you get unusual difficulty in breathing either during or after physical activity?”, was to identify those with either critical lung or coronary artery disease with likelihood of collapse during PA.

The sixth question, “Do you get palpitations either during or after physical activity?”, was for those who felt their heart beating faster than with similar previous PA. Palpitations had previously been identified as one of the symptoms felt by some who collapsed during PA [15].

The seventh question, “Do you get dizzy spells or fainting episodes either during or after physical activity?”, was owing to similar previous reports [17].

The primary objective of this study was to determine the level of difficulty in understanding of the SPARQ questions by members of the public. Secondary objectives were to determine the rate of persons declared unfit, if using the tool, and hence identify adults requiring further medical evaluation prior to participating in moderate or intensive PA, and obtain feedback on these questions.

## 2. Methods

This was a cross-sectional study on the use of the SPARQ. It was conducted as an anonymized public survey over a three-month period via an online platform, SoGo Survey, marketed by qualtrics^TM^, Seattle, WA, USA. Participation was voluntary and open to any member of the public 21 years of age and above. Randomness in volunteer participation could not be avoided in this survey. However, to minimize volunteer bias we reached out through various channels like governmental agencies, community grassroot and national sports agencies, professional societies and institutes of higher learning to help broaden the participation pool. In addition, to better ensure anonymity and confidentiality we reassured potential participants that their identities and data would be kept confidential.

Prior to survey participation, respondents were provided with basic information regarding the need for a self-administered pre-participation screening tool to identify persons at greater risk of adverse events during PA. Once formal consent was obtained, participants completed the three sections of the survey.

One section collected demographic information (age, sex, height, weight, ethnic group, smoking history and frequency of PA). The second was the SPARQ itself. The third was a survey on the difficulty of understanding each of the seven questions, the use of technical terms and the length of the questionnaire. Feedback was on a 5-point Likert scale. Participants were also asked for comments and suggestions on information included in the questionnaire, and their preferred frequency for completing the SPARQ. All survey responses were anonymized. Only study team members were given authorization to access the data.

Three participant sub-groups were created based on age. This study was powered to show a 10% difference in mean values between these sub-groups with a standard deviation of 25% using α = 0.50 and *p* = 0.05. This required a minimum number of 293 participants for each age-group in this study. Additionally, an incomplete survey response of at least 10% was expected. Hence, the minimum number was multiplied by 3 (for a total of 879) and raised to 1000.

This study was approved by the Institution Review Board of the Singapore Sports Institute (Approval Number SSG-EXP-002).

## 3. Data Analysis

The collected data were anonymized and automatically compiled into a Microsoft Excel database. Statistical analysis was carried out using IBM Corp. (Released 2012). IBM SPSS Statistics for Windows, Version 21.0. Armonk, NY: IBM Corp.

To conduct comparative analyses, respondents were categorized into three age groups (21–39.9 years, 40–59.9 years and ≥60 years). Respondents’ BMI and weekly PA were also grouped into five BMI categories (according to the WHO-proposed Asian BMI scale) and four PA groups (<150 min, 150–299 min, 300–449 min and ≥450 min per week) [18].

Descriptive statistics were applied to summarize nominal and ordinal data. Non-parametric comparisons of independent samples were conducted using the Kruskal–Wallis test, Mann–Whitney U test, and Chi-square tests, as appropriate. Parametric data comparisons were performed using *t*-tests and ANOVA. Reliability analysis (Cronbach’s alpha) was performed for the performance of the Likert scale in assessing ease of understanding of the questions.

### Feedback on SPARQ Data

In reviewing the responses from study participants, the authors considered satisficing in potentially influencing feedback results. Five types of satisficing behaviors were looked for amongst the survey respondents, viz. primary bias, acquiescence bias, early termination, non-response and straight-lining. Satisficing analysis was performed using methods suggested by Barge, Vriesema and Gehlbach [19,20]. The apparently satisficed data were removed from the feedback analysis and, owing to lack of universally-agreed definitions of difficulty in understanding when attempting such questions, the means of the Likert scores were arbitrarily divided into four groups of levels of difficulty as shown below:


**Difficulty Likert Index ***

**Level of Difficulty**
≥4.51No Difficulty4.01–4.50Slight Difficulty3.51–4.00Moderate Difficulty≤3.50Significant Difficulty* Assuming a 90% Likert 5 rate for no difficulty, 70% for slight difficulty, 50% for moderate difficulty and 30% for significant difficulty and the other responses equally distributed amongst the lower Likert numbers, the mean Likert scores derived were 4.75 (no difficulty), 4.25 (slight difficulty), 3.75 (moderate difficulty) and 3.25 (slight difficulty). Using these mean values, a scale of ≥4.51 to 5.00 to signify no difficulty, 4.01 to 4.50 to signify slight difficulty, 3.51 to 4.00 to signify moderate difficulty and ≤3.50 to signify significant difficulty was created.

Likert scale reliability analysis (Cronbach’s alpha) was determined for this survey. Free-text comments were also collected and grouped for possible feedback on future refinements of the questions.

## 4. Results

A total of 1137 persons participated in this study. One was under 21 years of age. No participants needed to be excluded from analysis for extensively incomplete data. Analysis was based on the remaining 1136 respondents.

### 4.1. Participant Characteristics (Table 1)

Males formed 56.5% of respondents and were older than females. The ethnic breakdown of the participants was generally similar to the population of Singapore, though there was slight under-representation of the elderly (19% vs. national 26.4%) and non-Chinese groups (Malays 5.9% vs. 12.8% and Indians 7.0% vs. 8.7%). The BMI was lower in the 21–39.9 and >60 years age-groups compared to those between 40–59.9 years age (24.2 and 23.6 vs. 24.7, *p* = 0.001). About 23.8% had at least one illness. Of these 18.5% had only one co-morbidity. Overall, 9.0% had hypertension, 4.8% dyslipidemia, 2.2% heart disease, 0.4% cancer, 3.3% diabetes mellitus and 9.7% had other illnesses. Persons with known medical illnesses were, generally older than those without.

Only 52.6% of respondents reported exercising for at least 150 min per week. Those >60 years of age spent more time on PA (318 min) versus 208 min for those between 40–59.9 years and 174 min for those <40 years age (*p* = 0.0001). There was no difference in the duration of PA per week for those with one or more forms of medical illness versus those without (*p* = 0.120). There was no decrease in PA duration with increasing numbers of co-morbidities (*p* = 0.862). Amongst persons with medical conditions, only those with cancer (median 150 vs. 60 min, *p* = 0.044) and those with other illnesses (median 160 vs. 120 min, *p* = 0.019) exercised less than those without such comorbidities.

Overall, 0.9% of participants stated that their medical conditions were not under control and 2.4% that the illness affected their ability to do PA.

### 4.2. Participants’ Responses to the Seven SPARQ Questions (Table 2)

The Likert scale reliability analysis performed for this survey showed a Cronbach’s alpha of 0.738.

Question 1 (Do you have either high blood pressure or a heart condition for which you still require treatment and close follow-up by a doctor?)

For this first question, 13.2% reported either high blood pressure or heart disease still requiring treatment or close follow-up by a doctor. These were slightly older than those without heart disease or hypertension. There was no difference in their weekly duration of PA. Moderate difficulty in understanding the question was encountered by those answering Yes. Respondents fed-back that the question covered two medical conditions, viz. high blood pressure and heart disease, and was, therefore, double-barreled.

Question 2 (Do you have moderate or severe joint pains or back pains made worse by physical exercise?)

For this question, 16.9% reported Yes. There was no significant age difference between those with or without such pains. Those with pain exercised less and encountered moderate difficulty with this question. For the 47 (4.1%) who expressed difficulty with this question, the terms moderate and slight were subjective and unsure whether back pain and backache were the same condition.

Question 3 (Have you been feeling unwell over the last one week with either fever, sore throat, cough, vomiting or diarrhea?)

For this question, 3.1% said Yes. There was no significant difference in age or usual duration of exercise with those who felt well. Those with symptoms expressed slight difficulty in understanding this question, while the others expressed no difficulty. Some felt that fever, cough, sore throat, vomiting and diarrhea should have been enquired about separately since persons may not likely have all these conditions at the same time.

Question 4 (Do you get chest pains either during or after physical activity?)

For this question, 2.5% answered Yes. There was no significant difference in age or numbers of cardiac risk factors from those not experiencing chest pain during PA. Those with symptoms generally exercised less and had moderate difficulty in understanding the question. The others did not encounter such difficulty. Those claiming difficulty were trying to grapple between what constituted chest pain and chest ache and wanted examples for better understanding.

Question 5 (Do you get unusual difficulty in breathing either during or after physical activity?)

Only 4.1% of participants experienced breathing difficulty that was unusual for that level of exercise. While there was no significant age difference amongst the groups, those with breathing difficulty exercised significantly less and experienced moderate difficulty in understanding the question. They felt that the word “unusual” was subjective and sometimes ambiguous. People who were exercising would always feel more breathless and wanted another word or phrase to better describe the abnormal symptoms.

Question 6 (Do you get palpitations either during or after physical activity?)

Up to 5.6% answered Yes. There was no significant age difference between those with or without palpitations. The symptomatic participants exercised less. Significant difficulty amongst symptomatic participants in understanding this question and moderate difficulty by others was because the word “palpitations” was considered too technical, and “medical jargon”, requiring simplification for better understanding.

Question 7 (Do you get dizzy spells or fainting episodes either during or after physical activity?)

For this question 6.1% answered “Yes”. Those with symptoms were generally younger, exercised much less and experienced slight difficulty in understanding. Those without symptoms had no difficulty. A few requested that the word “immediately” be added just before the words “after physical activity”. Some wanted the terms dizzy spells and fainting to be separate questions.

### 4.3. Participants Requiring Further Referral

Even though only 23.7% of the participants had a known prior medical condition, 35.7% of participants answered “Yes” to one or more of the seven questions posed in the SPARQ. For patients without known medical illnesses, 27.0% said Yes to one or more of the questions and would have been initially declared unfit for PA requiring further evaluation by a doctor versus 62.2% for those with a known medical illness (*p* = 0.000). When distributed by age this would be 32.0% of those from age 21–39.9 years, 36.0% of those from 40–59.9 years, and 38.0% of those aged 60 years and above (*p* = 0.277).

### 4.4. Participants Feedback on the SPARQ

#### Level of Difficulty in Understanding the SPARQ Questions

Generally, 90.0% of survey participants felt the quality of the advice provided in the SPARQ was adequate and 93.1% would be happy to follow it. The length of the questionnaire and the use of technical terms were felt to be adequate by 93.4% of the participants, regardless of presence of symptoms or co-morbidities. While overall satisfaction was high, targeted feedback revealed specific terminology concerns. Comments were received from 26.6% of survey participants. This included 28.8% from those with and 25.3% without symptoms.

A high rate of satisficing was observed (688 or 60.6%). The practice of satisficing was greater amongst those without symptoms (64.8%) versus 52.8% amongst those with symptoms (*p* = 0.000). The level of difficulty in understanding the questions was greater amongst those with co-morbidities for the first six questions. For Question 7 on dizzy spells and fainting no significant difference in level of understanding between those with or without symptoms was noted.

Many participants commented that the questions were well drafted and simple to understand. The suggestions and comments by the participants on these questions are summarized in Table 3. These comments and suggestions are to be utilized when considering refinements of the questions.

### 4.5. Suggested Frequency of SPARQ Usage

While only 40.6% of participants preferred the SPARQ to be used by exercise participants once annually, 36.6% preferred it be performed once every six months and the remaining 22.8% as frequently as needed. Those in older age groups preferred a once yearly schedule compared to those in younger age groups (*p* = 0.038).

## 5. Discussion

This study’s objectives were to determine the level of difficulty in understanding the questions in the SPARQ, the likelihood of potential exercise participants being declared unfit and requiring further medical evaluation, and to obtain feedback from the public on the document. The participants came from a broad age-spectrum and different walks of life. This is the first pre-participation tool co-designed with public feedback and also meant to address the need for those exercising in heat-prone environments that pose a higher risk of injury. Such feedback should be relevant for adult populations working and living in similar environments. This study was able to determine such difficulty in understanding the SPARQ questions. Thus, even though the level of satisfaction of the questionnaire exceeded 90%, 26.6% of participants provided comments. Most of these comments were not criticisms but provided for consideration of further improvements in the wording of the questionnaire.

Though the proportion of co-morbidities were expectedly higher in the older age groups with more of them requiring medical referral prior to actively engaging in PA, this, generally, did not result in less time spent by them on PA. This could have been because respondents in the younger age groups may have been actively employed with less time to spend on physical activities. This also suggests that the older age groups, though more prone to disease, may be more likely to engage in PA and therefore having a greater need for pre-participation screening. That those with co-morbidities had greater difficulty in understanding the questions also suggests that the construct of the questions needs to be addressed adequately to better ensure that those more prone to illnesses have less issues with using such questionnaires. This can be due to various factors including the complexity of the information, the patient’s cognitive or physical state, and their level of health literacy.

With the draft SPARQ, approximately one-third would have been declared unfit to engage in moderate or intense PA and be referred to a doctor for further evaluation. This was similar to the what would have occurred in a similar population using the Canadian GAQ, but less than would have been seen with the PAR-Q+ [15,17]. The majority of referrals were from questions 1 and 2, which may have been phrased in general terms, without adequate differentiation to the degree of active symptom control. There would be a need to review these two questions to determine whether further refinement of these may help decrease unnecessary referrals. After all, only 0.9% of participants stated that their medical conditions were not under control and only 2.4% that their illness affected the ability to do PA. Such refinement may also be considered for questions 4, 5 and 6. Naturally, there is concern as to whether the large numbers of potential referrals may unduly stress health resources. If one-third of all adults undergoing pre-participation screening need medical referral for further evaluation and testing, the numbers of doctors required to conduct such evaluations and the cost of these can be staggering [3]. For those whose symptoms or illnesses are well controlled, there should not be a need for further medical referral before continuing PA or gradually increasing their level of PA. Further referral may become a major disincentive to engage in PA, especially for the elderly. For reference, the use of the PARQ+ in the USA had seen much higher rates of referral in excess of 55%, even after appropriate modifications made [21]. That there was no significant difference in referral rates across age groups suggests that careful refining of the questions to identify those whose conditions are poorly controlled may better help determine the relationship between age and need for further medical referral. Just going by the duration of PA in the different age groups, it would not be far-fetched to consider whether the younger cohort may more likely also have poorly controlled disease that contributes to their lesser PA duration and therefore requiring medical referral. Therefore, careful wording of the questions may better help identify those requiring such referral.

Active ischemic heart disease, whether known or unknown, increases the risk of cardiovascular collapse during acutely stressful events such as unaccustomed PA, especially if not well controlled [1,14,22]. Therefore, the question on heart disease needs to be retained and refined by language that highlights lack of adequate control. A suggested revised question would be “Do you have heart disease that is not well controlled (i.e., chest pain, chest ache, chest tightness, chest discomfort and shortness of breath, in spite of medication) over the last six months?”

The role of hypertension vis-à-vis PA needs careful evaluation. Though hypertension is a known risk factor for coronary artery disease, adverse events had not been reported if the baseline blood pressure was 140–180 mm Hg systolic and ≤90 mm Hg diastolic [1]. The European Society of Cardiology 2020 guidelines recommend withholding maximal exercise stress testing if the BP exceeds 160/90 mm Hg [22]. Otherwise, if medically stable and with BP < 160/90 mm Hg further medical referral may not be required. The ACSM recommends that asymptomatic individuals with cardiovascular, metabolic or renal disease can continue exercising with progressive intensity unless new symptoms develop and that only patients with known uncontrolled BP (resting BP ≥ 140/90 mm Hg), stage 2 hypertension (BP ≥ 160/100 mm Hg) or target organ disease need undergo medical clearance before PA [3,23,24]. Therefore, only symptomatic, hypertensive patients or those with known uncontrolled high blood pressure may require medical evaluation before engaging in PA [25]. The question on hypertension may, thus, be revised to “Do you have difficulty in controlling your high blood pressure (having headaches with dizziness and blurring of vision) with medicines over the last six months?”

For question 2 on joint pains, a relatively common complaint, regular low-impact PA can maintain physical function and reduce pain levels in patients with osteoarthritis [26]. Understanding the International Olympic Committee’s concerns for musculoskeletal screening and increased risks of reinjury and contralateral side injury with recent ankle sprains, the language of this question needs refinement to minimise misunderstanding and reduce unnecessary referrals [27,28]. A possible revised question may be as follows: “Do you have aches and pains in your bones, joints and muscles that severely limits ability to perform physical exercise?”

Short, acute illnesses can predispose to heat injuries [29]. Heat disorders can occur with strenuous PA, especially when performed shortly after recent mild illnesses such as viral infections and gastroenteritis [30]. Many jurisdictions advise rest and avoidance of PA if recent such illnesses occur. Heat stroke can also result in cardiac complications including sudden cardiac death [31]. The very high understanding of this question suggests its retention. A slightly revised version of this question to ensure consistency in style with the other questions could be “Do you feel unwell over the last one week with either fever, sore throat, cough, vomiting or diarrhea?”

The small number of participants experiencing chest pains had moderate difficulty with this question. Chest pain during exertion suggests acute ischemic heart disease. Chest pain during exertion contributes to a high proportion of prodromal symptoms prior to collapse during PA [32,33,34]. Modifications to this question may allow better understanding. The need to avoid duplication with Question 1 that concerns ischemic heart disease should also be considered.

Unusual difficulty in breathing is a well-recognized prodromal symptom in previous studies on cardiac arrest during PA [21,35,36]. Unusual breathlessness may also portend a worsening respiratory condition [37]. The words, “unusual breathlessness” may require review. Such symptoms are relevant to the questionnaire. A suggested revised version may be “Do you feel more breathless when doing physical exercise than you usually would for that level of activity?”

The question on palpitations needs review. While palpitations may be a prodromal symptom in cardiac arrest patients, it has been equally prevalent in non-cardiac arrest patients in another study [38,39]. With significant misunderstanding issues and conflicting evidence of its value, one would need to review including this question in any pre-participation questionnaire.

There was only slight difficulty for the seventh question on dizzy spells or fainting episodes during or after PA. Dizziness is a prodromal symptom amongst cardiac arrest patients, including among athletes [17,39]. While the causes of dizziness may not be obvious initially it merits careful evaluation. The suggestion on adding the word “immediately” before “after physical activity” appears reasonable. The suggested revised version of this question would be “Have you felt dizzy or fainted either during or shortly after physical exercise over the last six months?”

Adopting and adapting the SPARQ as a primary pre-participation screening tool for PA may be considered with suitable modifications to the questions and after considering participants’ feedback. When applying such questionnaires across communities, large numbers of medical referrals may unduly strain existing medical resources [10]. While refinements may reduce referrals, this requires empirical verification.

## 6. Limitations of the Study

Even though Singapore has one of the highest digital literacy rates in the world and most homes have internet access, we recognize the challenges in accessing the survey for those without these. Our use of the online survey platform may have excluded a small segment of the population with low digital literacy or without internet access which may have resulted in some selection bias. We acknowledge the potential for randomness in volunteer participation and volunteer bias and its possible influence on this study’s findings. To minimize this, we reached out through various channels like governmental agencies, community grassroot and national sports agencies, professional societies and institutes of higher learning to help broaden the participation pool. In addition, to better ensure anonymity and confidentiality we re-assured potential participants that their identities and data will be kept confidential.

The ethnic composition of the study participants differed slightly from that of the country. This could be addressed in future such studies by use of stratified sampling.

This study could not examine the reliability of the SPARQ in minimizing the likelihood of adverse events during moderate to intense PA. While reliability of any pre-participation tool is important, predictive validity of such a questionnaire would have required a prospective long-term follow-up effort involving large numbers of community participants using this tool and monitoring of adverse outcomes after PA, which was beyond the scope of this study. In addition, none of the existing pre-participation questions that are commonly used have been known to have undergone a predictive reliability-testing process.

The presence of satisficing in this study was also a limitation as would be in similar questionnaires. However, efforts in identifying this helped to better understand the feedback from study participants. A high proportion of data were, unexpectedly, noted to be likely satisficed and removed from feedback analysis. This must be recognized as a likely limitation in surveys of this nature. Prior expectation of high satisficing rates would have resulted in the need to appropriately increase sample size. Measures were also applied to decrease satisficing behavior by trying to make the task of undertaking the survey easier and engaging through simplifying the task, trying to use clear and concise language and avoiding overly technical terms. In addition, a five-point, rather than a 7, 9 or 10-point Likert scale was used to minimize the number of choices and avoid complex question formats. The participant instructions were made fairly detailed so as to give respondents a clear understanding of the tasks and their purpose. The instructions had stated that their in-puts were valuable and can contribute to a larger goal. The survey was also kept relatively short so that it may be completed within ten minutes. In this anonymized public survey, we could not have clinically verified the presence/absence of actual conditions amongst those apparently unfit for PA. However, with actual individual use of the SPARQ, and further evaluation by a medical practitioner, if applicable, such verification and validation may have been performed.

In this study, the intensity of PA could not be defined owing to this being a survey-based study. Lay persons were not expected to be able to define the intensity of their PA in objective terms. Instead, duration of PA has been used, though this may not always equate to exercise intensity.

Finally, this questionnaire survey was not pilot-tested. Future studies on this should consider inclusion of such testing with cognitive interviews during tool development.

## 7. Conclusions

PA readiness questionnaires, such as the SPARQ, should be implemented community-wide, after due consultation with potential users, viz. members of the public, and also, perhaps, the sports community. Such an approach will more likely ensure that public safety is better served without having to send large sections of the community for medical evaluation and disincentivizing PA. All communities around the world may use the lessons from this study to consider validating their own such pre-participation tools. Of course, following such revision, there will be a need for significant efforts to implement and encourage use of the tool, even with digital and social media technology. These should allow and facilitate follow-up to assess the usefulness of such tools and empirically verify and validate their use for a safer sports environment.

## Figures and Tables

**Figure 1 healthcare-13-01837-f001:**
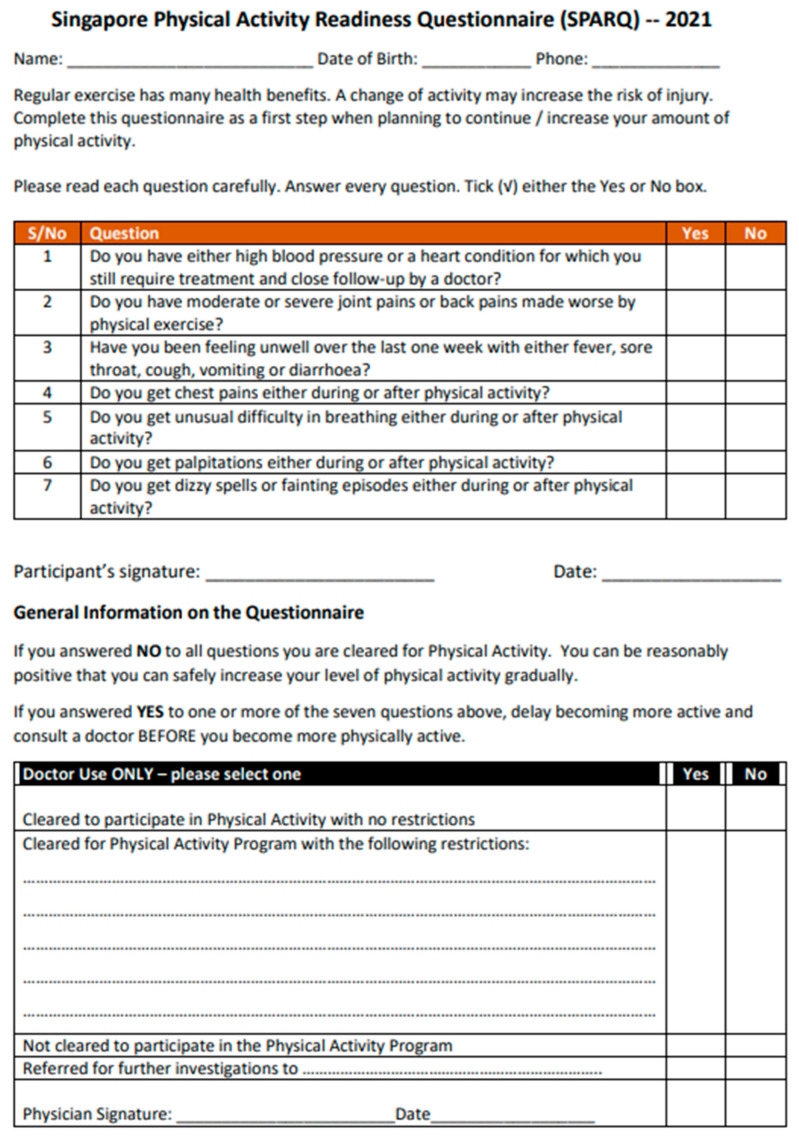
Singapore Physical Activity Readiness Questionnaire—2021.

**Table 1 healthcare-13-01837-t001:** Characteristics of the study participants.

Participant Characteristic	Data	Singapore Population Data
		(Where Available)
**Gender (%)**		
Male	642 (56.5%)	54.30%
Female	494 (43.5%)	45.70%
**Age (mean ± sd ^1^)**	48.7 ± 11.6 years	47.8 years
Males	49.5 ± 11.8 years	
Females	47.6 ± 11.3 years	
**Age Distribution**		
21–39.9 years	25.20%	35.60%
40–59.9 years	55.80%	38.00%
60–80.0 years	19.00%	26.40%
**Cigarette Smoker (%)**		
Yes	59 (5.2%)	NA
No	1077 (94.8%)	
**Basal Metabolic Index**		
Very Low (<18.5)	48 (4.2%)	NA
Low (18.5–22.9)	400 (35.2%)	
Moderate (23.0–27.4)	481 (42.3%)	
High (27.5–32.9)	156 (13.7%)	
Very High (>33.0)	51 (4.5%)	
**Ethnic Groups**		
Chinese	932 (82.0%)	75.30%
Malay	67 (5.9%)	12.80%
Indian	80 (7.0%)	8.70%
Others	57 (5.0%)	3.20%
**Duration of Physical Activity per week (minutes)**		NA
0–149 min	539 (47.4%)	NA
150–299 min	288 (25.3%)	
300–449 min	168 (14.8%)	
≥450 min	142 (12.5%)	
**Physical Activity per week (minutes) by age group**		
Age Group 21–39.9 years	174 ± 167	*p* < 0.0001
Age Group 40–59.9 years	208 ± 208	
Age Group > 60 years	314 ± 268	
**Physical Activity per week (minutes) by presence of co-morbidity**		
No co-morbidities	219 ± 205	*p* = 0.120
With co-morbidities	220 ± 253	
**Age Group of Respondents with co-morbidities**		
No co-morbidities = 866 (76.2%)	47.5 ± 11.3 years	*p* = 0.0001
With co-morbidities = 270 (23.8%)	52.5 ± 11.8 years	
Proportion of co-morbidities by age group:		
Age Group 21–39.9 years	45/286 (15.7%)	NA
Age Group 40–59.9 years	149/633 (23.5%)	
Age Group ≥ 60 years	76/216(35.2%)	
**By number of co-morbidities**		
1 co-morbidity	221/1136 (19.5%)	NA
2 co-morbidities	38/1136 (3.3%)	
3 or more co-morbidities	11/1136 (1.0%)	

^1^ sd = standard deviation.

**Table 2 healthcare-13-01837-t002:** Participants’ responses to the SPARQ.

S/No	Answering Yes/No to SPARQ ^1^ Question	Frequency, n (%)	Age in Years (*Mean* ± *SD* ^2^); *p* Value		Minutes per Week of PA ^3^ (*Mean* ± *SD* ^2^); *p* Value		Level of Understanding Index (*Mean* ± *SD* ^2^) for Non-Satisficing Participants		
							Difficulty Index	*p* Value	Level of Difficulty ^4^
**1.**	SPARQ ^1^ Question 1								
	a. Yes	150 (13.2%)	55.5 ± 10.8	*p* = 0.000	219.7 ± 226.1	*p* = 0.668	3.95 ± 1.11	*p* = 0.001	Moderate Slight
	b. No	985 (86.8%)	47.7 ± 11.4		219.2 ± 215.4		4.46 ± 0.87		
**2.**	SPARQ ^1^ Question 2								
	a. Yes	192 (16.9%)	49.3 ± 11.2	*p* = 0.442	193.7 ± 209.4	*p* = 0.007	3.72 ± 0.94	*p* = 0.0004	Moderate Slight
	b. No	944 (83.1%)	48.6 ± 11.7		224.5 ± 218.0		4.17 ± 0.91		
**3.**	SPARQ ^1^ Question 3								
	a. Yes	35 (3.1%)	47.6 ± 10.2	*p* = 0.582	188.9 ± 168.0	*p* = 0.491	4.19 ± 1.06	*p* = 0.003	Slight Nil
	b. No	1102 (96.9%)	48.7 ± 11.7		220.2 ± 218.1		4.70 ± 0.91		
**4.**	SPARQ ^1^ Question 4								
	a. Yes	28 (2.5%)	45.5 ± 12.7	*p* = 0.143	158.4 ± 236.5	*p* = 0.011	3.69 ± 1.08	*p* = 0.0003	Moderate Nil
	b. No	1109 (97.5%)	48.8 ± 11.6		220.8 ± 216.2		4.57 ± 0.78		
**5.**	SPARQ ^1^ Question 5								
	a. Yes	47 (4.1%)	45.9 ± 11.8	*p* = 0.088	148.8 ± 193.9	*p* = 0.0002	3.65 ± 1.23	*p* = 0.018	Moderate Slight
	b. No	1089 (95.9%)	48.8 ± 11.6		222.1 ± 217.3		4.17 ± 1.08		
**6**	SPARQ ^1^ Question 6								
	a. Yes	64 (5.6%)	46.2 ± 12.3	*p* = 0.072	142.7 ± 174.1	*p* = 0.0001	3.08 ± 1.36	*p* = 0.033	Significant Moderate
	b. No	1073 (94.4%)	48.9 ± 11.6		223.8 ± 218.3		3.54 ± 1.20		
**7.**	SPARQ ^1^ Question 7								
	a. Yes	69 (6.1%)	43.1 ± 10.1	*p* = 0.0004	141.7 ± 168.2	*p* = 0.0001	4.26 ± 1.06	*p* = 0.341	Slight Nil
	b. No	1067 (93.9%)	49.1 ± 11.7		224.4 ± 218.8		4.50 ± 0.81		

^1^ SPARQ = Singapore Physical Activity Readiness Questionnaire, ^2^ SD = Standard Deviation, ^3^ PA = Physical Activity, ^4^ Level of difficulty is defined as mean Likert scale index ≥ 4.51 (No Difficulty), 4.01–4.50 (Slight Difficulty), 3.51–4.00 (Moderate Difficulty), ≤3.50 (Significant Difficulty).

**Table 3 healthcare-13-01837-t003:** Comments, queries and suggestions by participants on each of the SPARQ questions.

**SPARQ Question**	**Common Comments ^1^**
1	Putting high blood pressure together with heart condition may be misleading. High blood pressure and heart condition, though interlinked, should be two different medical issues. High blood pressure is very common among middle age and seniors and if well controlled may not require medical referral. Some people do have high blood pressure but do not take their medications as prescribed by the doctor and think it does not require further treatment It is difficult to understand because of the word “still”. Still requiring treatment may not mean unstable. Have explanatory statements about terms, such as “heart condition” and “close follow-up”. Often people may not be aware that they have medical issues such as high blood pressure or heart disease and may answer No to this question.
2	Everyone will have a bit of joint pain or previous injuries and even slight high blood that does not need medical attention. Does that mean one cannot participate in any physical activities? Muscle ache may be mistaken for back pain. People who have recovered from old injuries and have occasional pains from exercising, might still want to engage in moderate or vigorous exercise. Different people will interpret moderate pain differently. Therefore, please provide examples and detailed explanation. Only some physical activities cause joint/back pains. Unsure whether to answer yes or no. When asking about severe or moderately severe joint points, it should be more specific, e.g., do you need to consult a doctor for joint pains? Would development of joint pains after exercising for a few months suggest unfitness to carry on exercising? There is a need to elaborate of various types and degrees of physical activity. Suggestion to add an explanation and further options if one were to click Yes, on any question. Suggestion to modify question to say “Do you currently have moderate or severe joint pains or back pains that become worse when you perform physical exercise?”. Why not just say joint pain or back pain without needing to weigh the degree of severity? Am not sure how useful this question will be. Use of objective pain indicators to describe mild, moderate or severe pain will help increase understanding of the question. Generally, people will get muscle pains from exercise
3	This question is easy to understand, but why list all these different complaints together. Suggest separating these different complaints as individual questions. A one-week period may be too long and counterproductive to physical fitness. Please consider reducing to just three days.
4	For chest pain, may be helpful to state what it means—is chest muscle ache also chest pain? Not sure about degree of chest pain before deciding to seek medical attention. A term like chest pain should be followed by everyday examples to help people identify what is normal and what is abnormal. Not sure how to define chest pains. Sometimes it feels like muscle strain? Then you would go to the doctor and do a ECG but it is just a muscle pull. The question needs layman’s explanation for clarity because it is quite normal to have chest discomfort during strenuous exercise like jogging.
5	The words “unusual difficulty in breathing” needs further explanation. People always get more breathless when exercising. Breathing difficulties are quite subjective and not so straightforward to distinguish clearly as yes/no. People may wonder what is meant by “unusual”. Needs to be clearly defined. Very vigorous exercise can cause a brief period of very hard breathing. Please be more specific about what is meant by unusual difficulty in breathing. Difficulty breathing will do, no need for ‘unusual’.
6	The word “palpitations” is medical jargon and not easily understood. Need to define in lay terms and give examples. Palpitations are quite subjective and not so straightforward to distinguish clearly as yes/no Suggest replacing ‘palpitations’ with simpler description. Maybe heart palpitations, irregular heartbeat, abnormal heart beats, fast heartbeat or feel heart is pounding? Does everyone understand what palpitations are? Not sure if “palpitation” is commonly used. It may be good to put down a short description. Palpitations are common when doing high-intensity exercises. The question requires more detailed explanation. Maybe find a simpler word for palpitations. “Palpitations” may not be easily understood by people. And most people get increased heart rate during/after vigorous exercise and they may not know how to differentiate regular increased heart rate and palpitations. This question can be better phrased. The word “palpitations” is difficult to understand. Include a definition in brackets. You want to be more specific and ask if physical activity causes heart palpitations.
7	Need to add the word “immediately” just before “after physical activity”. Please consider asking the questions on dizziness and fainting as separate questions rather than combining into a single question. Q 7 is difficult to give an answer because the question is not specific enough. Sometimes people don’t eat enough and may experience hypoglycemia. The question needed to be more specific. If can provide examples will be good.
Additional suggestions/comments/questions	The use of the word “either” can be removed from both of the questions of breath, before any moderate or intensive physical activity? Consider questions such as: are you currently vaccinated or been infected with COVID-19 before? Why did not include whether fully vaccinated? Travel overseas recently, name the country and date of return. Close contract of infected person recently? Date if yes. Did you had any operation performed over last one year? Maybe good to ask if you have performed warm up before exercise. To provide examples of moderate/vigorous exercises. Use of pictorials to describe the questions will greatly help on the explanation of the questions. What is meant by moderate and vigorous exercise needs to be clearly defined. Is brisk walking considered a moderate exercise? What about cycling? Or table tennis?” It may make more sense if the questions considered a specific time period (e.g., past 6 months) and frequency (a one-off event vs. a regular event over a 6-month period may have more risks) Rather than ‘do you get’, you should ask ‘have you ever experienced’ and for those people who answer ‘yes’ to follow up when the incident happens. If the questionnaire is on-line, can the final result and advice be auto-populated so as to be relevant to all who engage in physical activity. Kindly emphasize warming up before physical activities as top priorities and explain why. If questionnaire is to be on-line, allow for key words to be linked to explanations, if needed. Have you been advised to abstain from exercising before? If so, what was the reason? The questionnaire should advise the person to check with the doctor before exercise, especially if there have been previous injuries. Please define the term Physical Activity based on empirical research across the sector so that the community and industry can be clear and aligned when drafting advisories on this topic. Maybe can add in that if they feel unwell at any point. People should start exercising gradually. If you have stopped strenuous exercise for a while, you should ramp up the intensity gradually. Questionnaires for seniors should be slightly different Maybe can ask how the person is feeling on the day itself. Add advise of immediately stop exercising if one has those symptoms

^1^ Each of the Common Comments refer to those made by at least 10 survey participants.

## Data Availability

The raw data which form the basis for the Tables in this manuscript are available on request to the corresponding author.

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
