# Peer review of "The Singapore Physical Activity Readiness Questionnaire 2021 (SPARQ 2021)—Results of Public Feedback"

_healthcare, 2025, doi:10.3390/healthcare13151837_

Round 1
Reviewer 1 Report
Comments and Suggestions for Authors
Major Revisions
1 Terminology & Critical Errors
Correct "vaginal symptoms" (p.2) to "anginal symptoms" and "heartelated injuries" to "heat-related injuries". Standardize capitalization: "World Health Organization" (not Organisation), "physical activity" (lowercase unless sentence-initial).
2 Methodological Rigor
Address sampling bias: Acknowledge underrepresentation of elderly (19% vs. national 26.4%) and non-Chinese groups (Malay 5.9% vs. 12.8%). Propose stratified sampling in future validation studies.
Clinical validation gap: Add: "Self-reported comorbidities and 'unfit' status were not clinically verified – a limitation for clinical translation."
Pilot testing: Report any pilot phases or add: "Future studies should include cognitive interviews during tool development."
3 Tables & Figures
Table 2: Add footnotes defining "Level of Difficulty" thresholds.
Table 3: Group feedback thematically (e.g., Jargon, Ambiguity, Structural Issues). Remove redundant comments.
Figure 1: Ensure full visibility of doctor section. Increase font size for disclaimers (e.g., "delay becoming more active...").
Results & Discussion
Resolve contradictions: Clarify why 93.4% found terms "adequate" despite 26.6% submitting criticisms (e.g., "While overall satisfaction was high, targeted feedback revealed specific terminology concerns").
Clinical implications: Temper referral-reduction claims: "While refinements may reduce referrals, this requires empirical verification." Cite evidence: Link to studies showing tool refinements lowering referrals (e.g., reference [10]'s 41% reduction).
SPARQ's novelty: Emphasize: "This is the first pre-participation tool co-designed with public feedback for tropical contexts."
Minor But Essential Edits
Statistics: Report exact *p*-values (e.g., *p* = 0.027, not "p=0.000"). Add PA duration data to Table 1.
Grammar: Fix article omissions: "The ACSM recommends..." (p.11). Replace slash notation: "Mean ± SD; p-value" (not "Mean ± SD / p value").
Abbreviations: Define all acronyms at first use (e.g., "GAQ" in Introduction).
Author Response
Reviewer 1 Comments and Responses
Comment 1: Correct "vaginal symptoms" (p.2) to "anginal symptoms" and "heartelated injuries" to "heat-related injuries". Standardize capitalization: "World Health Organization" (not Organisation), "physical activity" (lowercase unless sentence-initial).
Response 1: Thank you for mentioning these. We have checked with the submitted manuscript. We could not find the words “vaginal symptoms” and “heartelated” in page 2. The words there are “anginal symptoms” and “heat-related injuries”. We have changed the spelling of the word “Organisation” to “Organization” as requested. We have also used the lower-case for “physical activity” unless sentence-initial.
Comment 2: Address sampling bias: Acknowledge underrepresentation of elderly (19% vs. national 26.4%) and non-Chinese groups (Malay 5.9% vs. 12.8%). Propose stratified sampling in future validation studies.
Response 2: Thank you for bringing this to our attention. We have acknowledged representation of the elderly and non-Chinese groups. The revised sentence is as follows “The ethnic breakdown of the participants was generally similar to the population of Singapore, though there was slight under-representation of the elderly (19% vs. national 26.4%) and non-Chinese groups (Malays 5.9% vs. 12.8% and Indians 7.0% vs. 8.7%).”
In addition, we have added a statement in the section on “Limitations” as follows: “The ethnic composition of the study participants differed slightly from that of the country. This could be addressed in future such studies by use of stratified sampling.”
Comment 3: Clinical validation gap: Add: "Self-reported comorbidities and 'unfit' status were not clinically verified – a limitation for clinical translation."
Response 3: Thank you for this comment. We had provided a paragraph in the section on “Limitations” which was as follows: “In this anonymized public survey, we could not have clinically verified the presence / absence of actual conditions amongst those apparently unfit for PA. However, with actual individual use of the SPARQ, and further evaluation by a medical practitioner, if applicable, such verification and validation may have been done.” We trust this paragraph would have addressed your comment on this matter.
Comment 4: Pilot testing: Report any pilot phases or add: "Future studies should include cognitive interviews during tool development."
Response 4: Thank you for bringing up this point. We have added a statement in the final paragraph of the section on “Limitations” to read as follows: “Finally, this questionnaire survey was not pilot-tested. Future studies on this should consider inclusion of such testing with cognitive interviews during tool development.”
Comment 5: Tables & Figures
Table 2: Add footnotes defining "Level of Difficulty" thresholds.
Table 3: Group feedback thematically (e.g., Jargon, Ambiguity, Structural Issues). Remove redundant comments.
Figure 1: Ensure full visibility of doctor section. Increase font size for disclaimers (e.g., "delay becoming more active...").
Response 5: We thank the reviewer for the comments on the Tables and Figures.
As requested, we have added footnotes defining Level of Difficulty thresholds to Table 2.
For Table 3, we had considered grouping the feedback comments and suggestions from the study participants in a thematic manner. While thematic grouping would have summarised such comments succinctly, this would have missed out the specifics of the comments for each question that would be helpful when considering and facilitating revisions to each of the questions. Grouping the comments thematically for each of the questions would have also appeared repetitive and lengthened the table further. Therefore, we had not grouped the comments and suggestions thematically. We trust the reviewer will understand the reasons for not adopting the thematic grouping approach for this table.
For Table 3, we have also made one major amendment, viz. left-indented all the items under Common Comments so as to make it more readable. We request that this amendment be maintained.
Regarding Figure 1, this was the document used by the study participants when the study was performed. The manuscript should reflect the actual document used. We will consider the reviewer’s suggestions on this in revisions to be made to the pre-participation questionnaire.
Comment 6: Results & Discussion
- Resolve contradictions: Clarify why 93.4% found terms "adequate" despite 26.6% submitting criticisms (e.g., "While overall satisfaction was high, targeted feedback revealed specific terminology concerns").
Response 6a: Thank you for making this valid point. We have expanded on this in the Results section, as suggested by the reviewer, by adding a statement and modifying another as follows: “While overall satisfaction was high, targeted feedback revealed specific terminology concerns. Comments were received from 26.6% of survey participants.” In addition, we had explained this in the Discussion section by adding the following statements “Thus, even though the level of satisfaction of the questionnaire exceeded 90%, 26.6% of participants provided comments. Most of these comments were not criticisms but provided for consideration of further improvements in the wording of the questionnaire.”
- Clinical implications: Temper referral-reduction claims: "While refinements may reduce referrals, this requires empirical verification."Cite evidence: Link to studies showing tool refinements lowering referrals (e.g., reference [10]'s 41% reduction).
Response 6b: Thank you for providing this advice. We have tempered the referral-reduction claims as suggested by the reviewer. We have moved these comments to the end of the discussion paragraph to include the reference citation and the statement so that these read as follows: “Adopting and adapting the SPARQ as a primary pre-participation screening tool for PA may be considered with suitable modifications to the questions and after considering participants’ feedback. When applying such questionnaires across communities, large numbers of medical referrals may unduly strain existing medical resources [10]. While refinements may reduce referrals, this requires empirical verification.”
- SPARQ's novelty: Emphasize: "This is the first pre-participation tool co-designed with public feedback for tropical contexts."
Response 6c: Thank you for the suggestion. We have modified the first paragraph of the discussion section to reflect this. The modifications are as follows: “This is the first pre-participation tool co-designed with public feedback and also meant to address the need for those exercising in heat-prone environments that pose a higher risk of injury. Such feedback should be relevant for adult populations working and living in similar environments.” This recognises that injuries during physical activity can occur in many environments and perhaps more so in heat-prone environments, such as tropical zones and also in temperate climates with high-intensity activities such as marathon runs that result in tremendous generation of heat.
Comment 7: Statistics: Report exact *p*-values (e.g., *p* = 0.027, not "p=0.000"). Add PA duration data to Table 1.
Response 7: Thank you for this advice. The p values that were reported as p=0.000 was because we had standardized the rounding of the numbers to three decimal places. As suggested by the reviewer, we have reported those that were originally given as p=0.000 in slightly greater detail, such as in four decimal places in Tables 1 and 2 or even one figure as p=<0.0001 in Table 1. Please advise whether the inconsistency is desirable or whether we should retain a consistent numbering system for p vales for up to three decimal places.
As requested by the reviewer, we have added the data on PA duration by age-group and by presence of co-morbidity to Table 1.
Comment 8: Grammar: Fix article omissions: "The ACSM recommends..." (p.11). Replace slash notation: "Mean ± SD; p-value" (not "Mean ± SD / p value").
Response 8: We thank the reviewer for the suggestion. We have amended the words “Mean ± SD / p value” to be in the format recommended by the reviewer wherever these occurred in the manuscript.
However, we are not so sure about the reviewer’s suggestion to italicize the word “The” before “ACSM” in page 12 and request for the Editorial Office’s input on this.
Comment 9: Abbreviations: Define all acronyms at first use (e.g., "GAQ" in Introduction).
Response 9: Thank you for the advice. We had already done so in the initial submission. Please advise whether and where we missed definitions of acronyms at first use in the manuscript.
Reviewer 2 Report
Comments and Suggestions for Authors
The manuscript has several keys for understanding the content:
There is a questionnaire proposal intending to diagnose the public who are presumed to engage in physical activity, and a semantic analysis.
There is a quantitative analysis focused on self-reported symptoms, and a qualitative, completely ignored database, even if it contains a few pertinent observations.
One of these suggestions, which I agree with, is defining the PA intensity, not only its duration. Physical effort may cause difficult breathing, increased heart rate or muscle pain, which may be confused with medical problems mentioned in the questionnaire.
A question can be raised about the questionnaire:
Was it validated, or is this study part of the process? Is the reliability important for this research instrument?
Author Response
Reviewer 2 Comments and Responses
Comment 1: There is a quantitative analysis focused on self-reported symptoms, and a qualitative, completely ignored database, even if it contains a few pertinent observations.
Response 1: Thank you for the comment. The intention of displaying the comments from the survey participants was to provide readers with the range of common items of public feedback that will enable subsequent revision of the questions, if found applicable and relevant. Obtaining feedback was also one of the secondary objectives of the study. Commenting on the relevance of each of the comments made may not have been a fruitful exercise, since the comments mentioned in Table 3 were common feedback comments from a wide variety of survey participants, comments which have to be considered by any jurisdiction creating such a questionnaire. We have added a statement in the Results section to that effect as follows: “These comments and suggestions are to be utilized when considering refinements of the questions.” We have added a statement in the first paragraph of the Discussion section that states “Most of these comments were not criticisms but provided for consideration of further improvements in the wording of the questionnaire.” We have also added a footnote to Table 3 to explain that each of those comments listed were made by at least 10 survey participants, since it would not have been possible to include every comment made by the survey participants.
Comment 2: One of these suggestions, which I agree with, is defining the PA intensity, not only its duration. Physical effort may cause difficult breathing, increased heart rate or muscle pain, which may be confused with medical problems mentioned in the questionnaire.
Response 2: Thank you for this comment. There were more than a thousand survey participants. They would have been challenged in defining the intensity of their physical activity in objective terms. Therefore, duration of physical exercise was used as a surrogate marker. To address this, we have added the following sentences to the section on Limitations as follows: “In this study, the intensity of PA could not be defined owing to this being a survey-based study and lay persons were not expected to be able to define the intensity of their PA in objective terms. Instead, duration of PA has been used, though this may not always equate to exercise intensity.”
Comment 3: A question can be raised about the questionnaire: Was it validated, or is this study part of the process? Is the reliability important for this research instrument?
Response 3: We thank the reviewer for this question. An objective of the study was to obtain public feedback on the questionnaire. That, by itself, would have also been an objective for any validation process. That more than 90% found the questionnaire acceptable is already one form of validation. Of course, with the feedback and comments obtained from the public, further refinements would be required to the seven questions that form the questionnaire. Such a revised version would be a better validated one. A similar concern would have been whether the questionnaire was pilot-tested. We have tried to address this concern by adding a statement to the limitations section as follows: “Finally, this questionnaire survey was not pilot-tested. Future studies on this should consider inclusion of such testing with cognitive interviews during tool development.” Then again, it may be argued that the study itself could have also been a form of pilot test for the SPARQ-2021 questionnaire. Similarly, the commonly-used such questionnaires around the world, such as the PAR-Q and the GAQ have not undergone rigorous pilot testing amongst members of the public or similar validation processes, but put out for use after consensus by a group of “content experts”.
Reviewer 3 Report
Comments and Suggestions for Authors
Manuscript ID: healthcare-3711493
Title: The Singapore Physical Activity Readiness Questionnaire 2021 (SPARQ 2021) – Results of Public Feedback
Journal: Healthcare
TITLE AND ABSTRACT
The title clearly reflects the content of the study. The abstract systematically summarizes the purpose, method, and main findings of the study. The finding that 35.7% of the 1136 participants would require medical evaluation before physical activity is clearly stated.
The clinical and public health importance and clinical relevance of the study can be stated more emphatically in the abstract. The tropical environmental conditions specific to Singapore and the reasons why the heat-related injury risks necessitated the development of SPARQ should be stated more clearly in the abstract. The special importance of Singapore's tropical climate conditions in terms of physical activity screening can be mentioned. In addition, addressing the limitations of the study in the abstract can make the article more balanced. More specific suggestions on practical applicability can be included in the conclusion section. Regionally important terms such as "Singapore", "tropical climate", "heat-related injuries" can be added to the keywords section. In addition, terms that will create a wider search area such as "screening", "public health", "exercise readiness" can be added.
INTRODUCTION
The introduction provides a strong theoretical foundation by presenting the benefits and risks of physical activity in a balanced manner. The topic is well-framed. The limitations of the previously used PAR-Q and GAQ questionnaires in Singapore are clearly stated, and the rationale for developing the SPARQ is logically explained. The scientific rationale and clinical basis for each SPARQ question are presented in detail. The rationale for developing the SPARQ and the specific needs of Singapore (e.g. hot and humid climate, population characteristics) are well-presented. Literature references (WHO, ACSM, PAR-Q+, GAQ) are sufficient and up-to-date. The WHO, ACSM and UK Chief Medical Officers physical activity recommendations are appropriately referenced.
More background information on the demographic characteristics and physical activity habits of the Singapore population could be provided. More scientific evidence supporting the effects of tropical climate conditions on physical activity safety could be provided A clear study hypothesis could be formulated. The research questions could be defined more clearly. It would be useful to emphasize the novelty of the SPARQ in the international context more clearly in the introduction. A more in-depth and critical comparison of the success/limitations of similar self-assessment tools in different populations would be useful in comparison with the existing literature. Furthermore, the heat-humidity link in the case of Singapore could be strengthened with more concrete case examples. Although physical activity modifications for Asian populations have been mentioned (reference 4), this topic should be expanded on. The impact of Singapore's unique climatic conditions, such as near 100% humidity and high temperatures, on physical activity risks should be explained in more detail. Furthermore, specific epidemiological data on the use of existing screening tools in the Singapore population should be presented.
METHODS
The cross-sectional survey method, sample size, subgroup analyses (age, BMI, PA, comorbidity) and statistical analyses (Kruskal-Wallis, t-test, ANOVA, etc.) of the study are explained in detail. The diversity of the participants (age, gender, ethnic distribution, etc.) and ethical approval processes are clearly stated. The study design is clearly defined. The sample size calculation was statistically appropriate (α = 0.05, 10% difference, 25% standard deviation). The separation into three age groups is reasonable. The use of an important methodological approach in survey studies, such as satisficing analysis, is one of the strengths of the study. Ethical approval was obtained and stated.
The selection criteria and strategies to prevent sampling bias of the online platform SoGo Survey are not sufficiently explained. Brief technical specifications (e.g. anonymity, data security) of the survey platforms used (SoGo Survey, Qualtrics) should be added. The use of the online survey platform may have excluded populations with low digital literacy or without internet access - this selection bias should be discussed. It should be explained why the difficulty levels determined for the Likert scale (>4.51 = No Difficulty, etc.) are categorized in this way. The inclusion and exclusion criteria for the study should be clearly stated. Possible biases in the selection and access of participants (e.g. social media, health institutions) should be stated. The Likert scale reliability analysis (Cronbach's alpha) in the survey should be reported clearly. The measures taken to evaluate and minimize the effects of voluntary participation bias should be explained. It may be stated whether a pilot study was conducted. In addition, the "satisficing" behavior and the method(s) for correcting it should be further explained, especially the limitations it brings should be clearly stated. In addition, a methodological assessment of the validity and reliability of the scales may be useful. These details are critical for the reproducibility of the study and the validity of the findings.
RESULTS
Participant characteristics are presented in comparison with Singapore population demographics, representativeness is assessed. Detailed analysis is performed for each SPARQ question, statistically assessing the relationships between age groups and physical activity duration. Levels of difficulty in understanding are classified using objective criteria, and specific feedback is categorized for each question. Table 2 and Table 3 provide comprehensive data presentation and support the findings. The 35.7% medical assessment rate is correctly highlighted as a clinically significant finding. Findings are presented systematically and in detail. Participant characteristics are presented in comparison with Singapore population data in Table 1. Responses to each SPARQ question are analyzed in terms of age, physical activity level, and difficulty in understanding. Table 2 includes comprehensive statistical analyses. Reporting of satisficing rates is valuable for methodological transparency.
The number and proportion of participants excluded from the satisficing analysis should be clearly stated. Practical significance should be discussed in comparisons between age groups. Confounding factors such as educational status and socioeconomic status of the participants were not analyzed. The reasons for not including them in the analysis can be explained. Concurrent validity was assessed, but predictive validity requires longitudinal follow-up, the reason for this should be explained and their inclusion should be considered. Multiple comparison corrections (e.g., Bonferroni) should be stated more clearly in subgroup analyses. The distribution of the 35.7% referral rate according to age groups was not found to be significant with p=0.277, but its clinical significance can be discussed. The relationship between the presence of comorbidity and difficulty in understanding SPARQ questions can be analyzed in more detail. The qualitative feedback in Table 3 can be presented more systematically with thematic analysis method. The titles of tables and figures can be more explanatory so that they can be understood on their own without the need to read the text. When presenting the findings in the text, instead of focusing only on statistical significance, more emphasis can be placed on what the findings mean in practical terms (effect size, interpretation of coefficients, etc.). For example, in addition to finding a regression coefficient to be "significant", it may be useful for the reader to interpret how much of a change a one-unit increase in the independent variable causes in the dependent variable.
SPARQ Question-Based Findings and Participant Feedback
For each question, the response rate, difficulty of meaning, and participant comments are reported in detail. Not only numerical analysis, but also qualitative needs are determined (e.g., which terms and expressions are incomprehensible or complex).
In response to problems in some feedback (e.g., technical terms, questions having "double" meanings), sample solution suggestions can be presented more clearly by the authors. In order to express the questions in a more objective/measurable manner, clinical or epidemiological sources can be cited and short examples can be given.
DISCUSSION
The clinical interpretation of the findings was done with a balanced and critical approach. The potential burden of the 35.7% medical assessment rate on the health system was objectively assessed and compared with previous studies. The discussion section compares the findings appropriately with the literature. The clinical significance of each SPARQ question and the revision suggestions were presented logically. Blood pressure threshold values ​​were discussed, especially with references to European and American guidelines on hypertension. Evidence-based recommendations were made with reference to ACSM and ESC guidelines for blood pressure management. The limitations of the study were stated honestly and the need for prospective validation studies was emphasized.
In the discussion, more focus could be given to the suggested changes (such as question examples, language simplification). In international comparison, the unique contribution of SPARQ and/or the model potential could be emphasized more strongly. In addition, the limitations of the study (“satisficing”, accuracy of survey responses, representativeness of the sample) should be examined in more depth and their possible impact on the results should be clearly stated. The interpretation of the findings should be further elaborated. Potential theoretical or practical explanations should be provided as to “why” these results occurred. Stronger connections can be made between the findings and existing theories. The relationship between Singapore's tropical climate conditions, especially with question 3 (symptoms of acute illness), has not been sufficiently discussed. The potential burden of the 35.7% referral rate on the health system should be supported by numerical data. The usability of SPARQ in other Asian countries and the need for cultural adaptation should be discussed. Recommendations for future research should be more specific. Recommendations for future research can be more specific and actionable.
CONCLUSIONS
The conclusion summarizes the main findings and indicates that the SPARQ can be used with appropriate modifications. The suggestion of rephrasing the questions to reduce unnecessary referrals is a practical approach. The aspects of the SPARQ that are suitable for use at the community level but need to be modified are clearly outlined. Recommendations for reducing the need for additional medical referrals are appropriate. The authors discuss the implications of the results for practical application and policy-makers.
This section should be more than just a summary. The study's potential contributions to public health policy should be stated. It should more concretely state the study's practical implications and implications for policy-makers, administrators, or clinicians in the health care field. It should conclude the paper strongly by emphasizing the study's "big picture" importance and its ultimate contribution to the field. Suggestions should be provided on how to validate the revised version of the SPARQ. The conclusion should be more specific, action-oriented; that is, it can be strengthened by direct suggestions on which questions can be rephrased and how. Additionally, insights can be added on how SPARQ can be adapted and tested in different countries and cultures. Specific action items and implementation timelines can be suggested. Digital implementation strategies and technology integration suggestions can be added. Policy implications should be evaluated in more detail. Specific methodology suggestions can be made for future validation studies.
Ethics and References
References are selected from current, relevant, and high-impact journals. Tables and figures are of sufficient quality and are self-explanatory. Ethical approval and conflict of interest declarations have been made as appropriate. Ethical approval has been obtained and stated. Informed consent processes have been explained. A conflict of interest declaration has been provided. Author contributions have been stated in detail.
No data sharing policy has been stated. Whether the study protocol has been previously registered (e.g. ClinicalTrials.gov) should be stated. More recent literature review can be done for some critical areas. The readability test results of the SPARQ questionnaire can be clearly stated in Figure 1. The statistical software version is stated but specific analysis packages are not. A data availability statement can be added.
Author Response
Reviewer 3 Comments and Responses
Comment 1. TITLE AND ABSTRACT
- The title clearly reflects the content of the study. The abstract systematically summarizes the purpose, method, and main findings of the study. The finding that 35.7% of the 1136 participants would require medical evaluation before physical activity is clearly stated.
Response 1a. Thank you
- The clinical and public health importance and clinical relevance of the study can be stated more emphatically in the abstract. The tropical environmental conditions specific to Singapore and the reasons why the heat-related injury risks necessitated the development of SPARQ should be stated more clearly in the abstract. The special importance of Singapore's tropical climate conditions in terms of physical activity screening can be mentioned.
Response 1b. Thank you for bring up this point. The heat-related injury risks in the tropical environmental conditions were only one of the factors that necessitated the development of SPARQ. Concerns about the complexity and difficulty in self-administering other pre-participation tools were the main factors and this has been mentioned in page 2 of the manuscript and in the abstract. We have, therefore, amended the second sentence of the abstract to read: “Concern over the usefulness and difficulty in understanding of these questions led to the design of a Singapore Physical Activity Readiness Questionnaire (SPARQ).” One needs to be mindful that heat-related injuries do not occur only in hot tropical environments such as Singapore, but also in temperate countries, and especially during intensive physical activity such as marathon runs in the area of sports. The reference to heat-related injuries was made because mention of this was minimal in other existing pre-participation questionnaires. Therefore, even though the study was done in Singapore, the results would be relevant to many other jurisdictions that lie not only in the tropics, but also in other parts of the world such as the USA, Britain, Japan and Australia. Therefore, we did not over-emphasize the role played by heat-related injuries, which, while being important, were not the only factors to be considered.
- In addition, addressing the limitations of the study in the abstract can make the article more balanced. More specific suggestions on practical applicability can be included in the conclusion section.
Response 1c. Thank you for the suggestion to include the limitations of the study in the abstract. We have considered this carefully. The inclusion of the various limitations (which are mainly methodological) would have unnecessarily confused the reader about the focus of the study, and would have severely increased the length of the abstract. We are aware of the word limit for the abstract and have carefully kept to this word limit. We hope the reviewer appreciates that only the principal aspects of the manuscript would be in the abstract. As of now, we would wish to retain the abstract, as is, except for the earlier changes made.
Even for the conclusion section in the abstract, the focus should be on the implications of public feedback on the SPARQ, which would be on the need to review it and revise it based on the feedback received rather than on other practical applications of the SPARQ. Therefore, the conclusion in the abstract remains unchanged.
- Regionally important terms such as "Singapore", "tropical climate", "heat-related injuries" can be added to the keywords section. In addition, terms that will create a wider search area such as "screening", "public health", "exercise readiness" can be added.
Response 1d. We thank the reviewer for the suggestions on additional key words. As explained earlier heat injuries and Singapore were not the prime foci of this study and manuscript. The term “screening” is relevant, though, and we have added this word to the list of key words.
Comment 2. INTRODUCTION
- The introduction provides a strong theoretical foundation by presenting the benefits and risks of physical activity in a balanced manner. The topic is well-framed. The limitations of the previously used PAR-Q and GAQ questionnaires in Singapore are clearly stated, and the rationale for developing the SPARQ is logically explained. The scientific rationale and clinical basis for each SPARQ question are presented in detail. The rationale for developing the SPARQ and the specific needs of Singapore (e.g. hot and humid climate, population characteristics) are well-presented. Literature references (WHO, ACSM, PAR-Q+, GAQ) are sufficient and up-to-date. The WHO, ACSM and UK Chief Medical Officers physical activity recommendations are appropriately referenced.
Response 2a. Thank you.
- More background information on the demographic characteristics and physical activity habits of the Singapore population could be provided. More scientific evidence supporting the effects of tropical climate conditions on physical activity safety could be provided.
Response 2b. We thank the reviewer for these suggestions. As mentioned in Response 1b earlier, the emphasis on this study was not only heat-related injuries and physical activity habits of the Singapore population. The Results of the study already throw some light on the physical characteristics and physical activity habits of the Singapore population. Expanding on these areas in the introduction would be likely to slant the focus of the study to a discussion on the sports habits of people in Singapore, which is clearly not the primary aim of this study and manuscript. In addition, at least three references (29, 30, 31) provide scientific evidence of the effects of heat-injury during strenuous sports. We would request that there should not be a need to expand on these areas which, we hope, have already been reasonably covered in this manuscript.
- A clear study hypothesis could be formulated. The research questions could be defined more clearly. It would be useful to emphasize the novelty of the SPARQ in the international context more clearly in the introduction. A more in-depth and critical comparison of the success/limitations of similar self-assessment tools in different populations would be useful in comparison with the existing literature. Furthermore, the heat-humidity link in the case of Singapore could be strengthened with more concrete case examples. Although physical activity modifications for Asian populations have been mentioned (reference 4), this topic should be expanded on. The impact of Singapore's unique climatic conditions, such as near 100% humidity and high temperatures, on physical activity risks should be explained in more detail. Furthermore, specific epidemiological data on the use of existing screening tools in the Singapore population should be presented.
Response 2c. We thank the reviewer for these various suggestions. The aims of the study (primary and secondary) were, as documented in the manuscript, to:
- determine the level of difficulty in understanding of the SPARQ questions by members of the public
- determine the rate of persons declared unfit, if using the tool, and hence identify adults requiring further medical evaluation prior to participating in moderate or intensive PA
- obtain feedback on these questions
That being the case there was no need to create a study hypothesis, nor draw up additional research questions. Since SPARQ had not yet been implemented in Singapore at the time of the study, which was meant to obtain public feedback before revisions and implementation, we are not in a position to emphasize the novelty of SPARQ in the international context in the section on Introduction.
The aim of the study was also not to compare SPARQ with other existing self-assessment tools. Such a comparison would have required a very different methodology and was not the intention of this study. As mentioned earlier we had already provided three other references (29, 30, 31) for heat injuries. Heat-related injuries were also not the main focus of this paper. We did not want to distract the reader from the focus of the study.
Comment 3. METHODS
- The cross-sectional survey method, sample size, subgroup analyses (age, BMI, PA, comorbidity) and statistical analyses (Kruskal-Wallis, t-test, ANOVA, etc.) of the study are explained in detail. The diversity of the participants (age, gender, ethnic distribution, etc.) and ethical approval processes are clearly stated. The study design is clearly defined. The sample size calculation was statistically appropriate (α = 0.05, 10% difference, 25% standard deviation). The separation into three age groups is reasonable. The use of an important methodological approach in survey studies, such as satisficing analysis, is one of the strengths of the study. Ethical approval was obtained and stated.
Response 3a. Thank you.
- The selection criteria and strategies to prevent sampling bias of the online platform SoGo Survey are not sufficiently explained.
Response 3b. Thank you for the remarks. The selection criteria are simple and mentioned in the first paragraph of methodology, viz. any member of the public of age 21 years and above. There were no other criteria used. To minimize sampling bias a wide spectrum of society was approached consisting of members of the public in various public organisations, educational institutions, sports organisations and community organisations. We have modified the first paragraph of the Methodology section to better address the reviewer’s concerns. The revised sentences are as follows: “Randomness in volunteer participation could not be avoided in this survey. However, to minimize volunteer bias we reached out through various channels like The study was publicised through governmental agencies, community grassroot and national sports agencies, professional societies and institutes of higher learning to help broaden the participation pool. In addition, to better ensure anonymity and confidentiality we re-assured potential participants that their identities and data will be kept confidential.”
The SoGo platform is simply an on-line platform for the conduct of surveys. In a country such as Singapore with one of the highest literacy rates in the world and with almost every home having internet access, the on-line platform would be most likely to garner a reasonably representative sample. Anyone with a computer and with internet access would have been able to access the SoGo platform, the link for which was available in the participant information sheet after formal consent to participate had been obtained so that anyone with a computer and internet access could use the platform. Therefore, the SoGo platform, by itself, would be unlikely to result in any sampling bias.
- Brief technical specifications (e.g. anonymity, data security) of the survey platforms used (SoGo Survey, Qualtrics) should be added. The use of the online survey platform may have excluded populations with low digital literacy or without internet access - this selection bias should be discussed.
Response 3c. Thank you for addressing these points. We have, as requested by the reviewer, added two sentences to the third paragraph in the Methodology section. They are as follows: “All survey responses were anonymized. Only study team members were given authorization to access the data.” These statements would address anonymity and data security concerns.
Regarding the matter of lack of internet access for potential participants, even though there is a relatively high digital literacy rate in Singapore and most homes in Singapore would have internet access, we have, as requested by the Reviewer, added a statement in the Limitations section on this matter as follows: “ Even though Singapore has one of the highest digital literacy rates in the world and most homes have internet access, we recognise the challenges in accessing the survey for those without these. Our use of the online survey platform may have excluded a small segment of the population with low digital literacy or without internet access and may have resulted in some selection bias.”
- It should be explained why the difficulty levels determined for the Likert scale (>4.51 = No Difficulty, etc.) are categorized in this way. The inclusion and exclusion criteria for the study should be clearly stated. Possible biases in the selection and access of participants (e.g. social media, health institutions) should be stated.
Response 3d. Thank you for bringing up this point. When responses on level of difficulty are recorded using a five-point Likert scale, there will likely be differences in difficulty level experienced by different participants. There are currently no internationally accepted criteria on categorization of difficulty levels based on the use of a five-point Likert scale. We arbitrarily assumed a 90% Likert 5 rate for No difficulty, 70% for slight difficulty, 50% for moderate difficulty and 30% for significant difficulty and the other responses equally distributed amongst the lower Likert numbers. With this approach we obtained the mean Likert scores derived as 4.75 (no difficulty), 4.25 (slight difficulty), 3.75 (moderate difficulty) and 3.25 (slight difficulty). Using these mean values we created a scale of ≥4.51 to 5.00 to signify no difficulty, 4.01 to 4.50 to signify slight difficulty, 3.51 to 4.00 to signify moderate difficulty and ≤3.50 to signify significant difficulty. The lack of universally accepted criteria has already been mentioned in the manuscript in the section on “Feedback on SPARQ data”. We have now added our calculation of the difficulty scale as a footnote to the short table on difficulty scales in this section. In addition, we have added the definition of these criteria as a footnote to Table 2.
We have added a note on possible biases in selection and access of participants in the section on Limitations.
- The Likert scale reliability analysis (Cronbach's alpha) in the survey should be reported clearly.
Response 3e: Thank you for the suggestion. We have done the Likert scale reliability analysis. The Cronbach’s alpha is 0.738. Determination of the Cronbach’s alpa has now been mentioned in the section of data analysis and in the results section.
- The measures taken to evaluate and minimize the effects of voluntary participation bias should be explained.
Response 3f: Thank you for this suggestion. We have added some statements in the Limitations section to acknowledge voluntary participation bias as follows: “We acknowledge the potential for randomness in volunteer participation and volunteer bias and its possible influence on the study's findings”.
In addition, we have added statements onmeasures taken to minimize this as follows: “To minimize this we reached out through various channels like governmental agencies, community grassroot and national sports agencies, professional societies and institutes of higher learning to help broaden the participation pool. In addition, to better ensure anonymity and confidentiality we re-assured potential participants that their identities and data will be kept confidential.”
- It may be stated whether a pilot study was conducted.
Response 3g: A pilot study was not conducted prior to the conduct of this survey. A statement on this has been included in the section on limitations as follows: “Finally, this questionnaire survey was not pilot-tested. Future studies on this should consider inclusion of such testing with cognitive interviews during tool development.”
- In addition, the "satisficing" behavior and the method(s) for correcting it should be further explained, especially the limitations it brings should be clearly stated. In addition, a methodological assessment of the validity and reliability of the scales may be useful. These details are critical for the reproducibility of the study and the validity of the findings.
Response 3h. Thank you for the suggestions. Being a study based on a public response survey, all participants were asked to state their responses as honestly as possible. The high satisficing rate was unexpected. We have expanded the paragraph on satisficing in the section on Limitations to read as follows: “A high proportion of data were, unexpectedly, noted to be likely satisficed and removed from feedback analysis. This must be recognized as a likely limitation in surveys of this nature. Prior expectation of high satisficing rates would have resulted in the need to appropriately increase sample size. Measures were also applied to decrease satisficing behaviour by trying to make the task of undertaking the survey easier and engaging by simplifying the task, trying to use clear and concise language and avoid overly technical terms. In addition, a five-point, rather than a 7, 9 or 10-point Likert scale was used to minimize the number of choices and avoid complex question formats. The participant instructions were made fairly detailed so as to give respondents a clear understanding of the tasks and their purpose. The instructions had stated that their in-puts were valuable and can contribute to a larger goal. The survey was also kept relatively short so that it may be completed within ten minutes.”
The issue of methodological assessment has been addressed in Response 3e.
Comment 4. RESULTS
- Participant characteristics are presented in comparison with Singapore population demographics, representativeness is assessed. Detailed analysis is performed for each SPARQ question, statistically assessing the relationships between age groups and physical activity duration. Levels of difficulty in understanding are classified using objective criteria, and specific feedback is categorized for each question. Table 2 and Table 3 provide comprehensive data presentation and support the findings. The 35.7% medical assessment rate is correctly highlighted as a clinically significant finding. Findings are presented systematically and in detail. Participant characteristics are presented in comparison with Singapore population data in Table 1. Responses to each SPARQ question are analyzed in terms of age, physical activity level, and difficulty in understanding. Table 2 includes comprehensive statistical analyses. Reporting of satisficing rates is valuable for methodological transparency.
Response 4a. We thank the reviewer for the comments.
- The number and proportion of participants excluded from the satisficing analysis should be clearly stated.
Response 4b. Thank you for this reminder. We had missed mentioning the number previously. We have added a statement in the Results section to address these, as follows: “A high rate of satisficing was observed (688 or 60.6%)”.
- Practical significance should be discussed in comparisons between age groups. Confounding factors such as educational status and socioeconomic status of the participants were not analyzed. The reasons for not including them in the analysis can be explained.
Response 4c. Thank you for this suggestion. We have added a paragraph in the Discussion to address this and is as follows: “Though the proportion of co-morbidities were expectedly higher in the older age groups with more of them requiring medical referral prior to actively engaging in PA, this, generally, did not result in less time spent by them on PA. This could have been because respondents in the younger age groups may have been actively employed with less time to spend on physical activities. This also suggests that the older age groups, though more prone to disease, may be more likely to engage in PA and therefore having a greater need for pre-participation screening. That those with co-morbidities had greater difficulty in understanding the questions also suggests that the construct of the questions need to be addressed adequately to better ensure that those more prone to illnesses have less issues with using such questionnaires.”
We did not look into educational status and socio-economic status in the survey. While these may have some correlation with difficulty in understanding the questions, the intent to simplify and shorten the survey questions would have helped persons both of higher or lower educational status or socio-economic status. Inclusion of questions to determine educational status and socio-economic status would have likely increased the number of questions in the study and affected participation rates and likely increased the likelihood of satisfying. The inclusion of these data would not likely have added value to the aims of the study.
- Concurrent validity was assessed, but predictive validity requires longitudinal follow-up, the reason for this should be explained and their inclusion should be considered. Multiple comparison corrections (e.g., Bonferroni) should be stated more clearly in subgroup analyses.
Response 4d. Thank you for mentioning this. We had made reference to this in the section on limitations. However, we have amended that portion to better address the point on predictive validity. The revised version reads as follows: “While reliability of any pre-participation tool is important, predictive validity of such a questionnaire would have required a prospective long-term follow-up effort involving large numbers of community participants using this tool and monitoring of adverse outcomes after PA, which was beyond the scope of this study. In addition, none of the existing pre-participation questions that are commonly used have been known to have under-gone a predictive reliability-testing process.”
Multiple comparison corrections were not made in sub-group analysis in this study.
- The distribution of the 35.7% referral rate according to age groups was not found to be significant with p=0.277, but its clinical significance can be discussed.
Response 4e. Thank you for the suggestion. In this survey, the referral rate was based purely on the respondents answering Yes or No to each of the seven questions. None of the seven questions considered whether the condition was poorly controlled. We have, thus, added the following statements into the discussion section: “That there was no significant difference in referral rates across age groups suggests that careful refining of the questions to identify those whose conditions are poorly controlled may have better helped determine the relationship between age and need for further medical referral. Just going by the duration of PA in the different age groups, it would not be far-fetched to consider whether the younger cohort may more likely also have poorly controlled disease that contributes to their lesser PA duration and therefore requiring medical referral. Therefore, careful wording of the questions may better help identify those requiring such referral.”
- The relationship between the presence of comorbidity and difficulty in understanding SPARQ questions can be analyzed in more detail.
Response 4f: We have added a sentence to the statements added in Response 4c to suggest reasons for the relationship between presence of comorbidity and difficulty in understanding the questions. The statement is: “This can be due to various factors including the complexity of the information, the pa-tient's cognitive or physical state, and their level of health literacy.”
- The qualitative feedback in Table 3 can be presented more systematically with thematic analysis method.
Response 4g: Thank you for this suggestion. For Table 3, we had considered grouping the feedback comments and suggestions from the study participants in a thematic manner. While thematic grouping would have summarised such comments succinctly, this would have likely missed out the specifics of the comments for each question that would be pertinent when considering and facilitating revisions to each of the questions. Grouping the comments thematically for each of the questions would have also appeared repetitive and lengthened the table further. Therefore, we had not grouped the comments and suggestions thematically. We trust the reviewer will understand the reasons for not adopting the thematic grouping approach for this table.
- The titles of tables and figures can be more explanatory so that they can be understood on their own without the need to read the text. When presenting the findings in the text, instead of focusing only on statistical significance, more emphasis can be placed on what the findings mean in practical terms (effect size, interpretation of coefficients, etc.). For example, in addition to finding a regression coefficient to be "significant", it may be useful for the reader to interpret how much of a change a one-unit increase in the independent variable causes in the dependent variable.
Response 4h: Thank you for the suggestions. We have reviewed the titles of the tables and figures. They are all short and concise and reflect the content in the tables. There would be no value in lengthening the titles. We have, however, deleted some of the words in Table 2 to reduce repetitiveness and improve the formatting of the table.
We have also considered the suggestion to describe how much of a change a one-unit increase in the independent variable causes in the dependent variable. We do not understand the suggestion nor the reason for it. In Table 2, the words significant do not mean significant difference. They refer to significant difficulty in understanding. We have provided an explanatory footnote at the bottom of Table 2 to describe the terms “Significant difficulty, Moderate difficulty, Slight difficulty and Nil difficulty. With the footnotes, a reader would be able to interpret the table on its own without needing to refer to the text.
Comment 5: SPARQ Question-Based Findings and Participant Feedback
- For each question, the response rate, difficulty of meaning, and participant comments are reported in detail. Not only numerical analysis, but also qualitative needs are determined (e.g., which terms and expressions are incomprehensible or complex).
Response 5a: Thank you
- In response to problems in some feedback (e.g., technical terms, questions having "double" meanings), sample solution suggestions can be presented more clearly by the authors. In order to express the questions in a more objective/measurable manner, clinical or epidemiological sources can be cited and short examples can be given.
Response 5b: Thank you for these suggestions. We have provided sample solution suggestions to some of these questions in the discussion section. The clinical bases for these, with sources mentioned, have already been provided in the text. The main issues with most of the questions that required revision were on clear understanding, simplicity in language and minimizing use of jargon or technical terms.
Comment 6: DISCUSSION
- The clinical interpretation of the findings was done with a balanced and critical approach. The potential burden of the 35.7% medical assessment rate on the health system was objectively assessed and compared with previous studies. The discussion section compares the findings appropriately with the literature. The clinical significance of each SPARQ question and the revision suggestions were presented logically. Blood pressure threshold values ​​were discussed, especially with references to European and American guidelines on hypertension. Evidence-based recommendations were made with reference to ACSM and ESC guidelines for blood pressure management. The limitations of the study were stated honestly and the need for prospective validation studies was emphasized.
Response 6a: Thank you.
- In the discussion, more focus could be given to the suggested changes (such as question examples, language simplification).
Response 6b: Thank you for the suggestion. We have added a sentence in the second paragraph of the Discussion section as follows: “Therefore, careful wording of the questions may better help identify those requiring such referral.” In addition, rewording of some of the SPARQ questions has been suggested in the Discussion section.
- In international comparison, the unique contribution of SPARQ and/or the model potential could be emphasized more strongly. In addition, the limitations of the study (“satisficing”, accuracy of survey responses, representativeness of the sample) should be examined in more depth and their possible impact on the results should be clearly stated.
Response 6c: Thank you for mentioning this. As mentioned earlier, we have added to the text that SPARQ is the first pre-participation tool co-designed with public feedback and that includes the need to identify those prone to heat-injuries. We have also expanded on a number of limitations of the study, including satisficing, accuracy of survey responses and representativeness of the sample, examining them in greater depth.
- The interpretation of the findings should be further elaborated. Potential theoretical or practical explanations should be provided as to “why” these results occurred. Stronger connections can be made between the findings and existing theories. The relationship between Singapore's tropical climate conditions, especially with question 3 (symptoms of acute illness), has not been sufficiently discussed.
Response 6d: Thank you for these comments. As explained earlier, explanations have been suggested for some of the results. This study is not only about Singapore’s climatic conditions and question 3. Heat injuries occur in both tropical and temperate environments. Their importance and relevance has been emphasized with three references. They need to be considered in pre-participation PA screening. Their inclusion is not just for Singapore, or for tropical countries or for Asia, but for all countries.
- The potential burden of the 35.7% referral rate on the health system should be supported by numerical data.
Response 6e: Thank you for suggesting this. We have added in the Discussion section that “If one-third of all adults undergoing pre-participation screening need medical referral for further evaluation and testing, the numbers of doctors required to conduct such evaluations and the cost of these can be staggering [3]”. We have added a reference to this statement. Numerical data would not be required to imagine the potential burden of this on the health system of any community. Once a person is referred for medical evaluation, a variety of tests would be required, such as electrocardiography, echocardiography and also cardiac stress testing. These are all expensive tests and require doctors and medical technologists for their conduct. Most countries may not have the numbers of such people or equipment resources to manage one third of their adult population with these assessments.
- The usability of SPARQ in other Asian countries and the need for cultural adaptation should be discussed.
Response 6f: Thank you for mentioning this. As explained earlier, SPARQ is not only for Singapore or for Asia. Its usability is universal. We have alluded to this in te manuscript.
- Recommendations for future research should be more specific. Recommendations for future research can be more specific and actionable.
Response 6g: Thank you for this suggestion. For the moment, until such a research project is clearly defined, it would not be appropriate to make it specific. We agree that it should be actionable.
Comment 7: CONCLUSIONS
- The conclusion summarizes the main findings and indicates that the SPARQ can be used with appropriate modifications. The suggestion of rephrasing the questions to reduce unnecessary referrals is a practical approach. The aspects of the SPARQ that are suitable for use at the community level but need to be modified are clearly outlined. Recommendations for reducing the need for additional medical referrals are appropriate. The authors discuss the implications of the results for practical application and policy-makers.
Response 7a: Thank you
- This section should be more than just a summary. The study's potential contributions to public health policy should be stated. It should more concretely state the study's practical implications and implications for policy-makers, administrators, or clinicians in the health care field. It should conclude the paper strongly by emphasizing the study's "big picture" importance and its ultimate contribution to the field. Suggestions should be provided on how to validate the revised version of the SPARQ. The conclusion should be more specific, action-oriented; that is, it can be strengthened by direct suggestions on which questions can be rephrased and how. Additionally, insights can be added on how SPARQ can be adapted and tested in different countries and cultures. Specific action items and implementation timelines can be suggested. Digital implementation strategies and technology integration suggestions can be added. Policy implications should be evaluated in more detail. Specific methodology suggestions can be made for future validation studies.
Response 7b: Thank you for the feedback. We have revised our conclusion based on these suggestions. The revised conclusion is as follows: “PA readiness questionnaires, such as SPARQ, should be implemented community-wide, after due consultation with potential users, viz. members of the public, and also, perhaps, the sports community. Such an approach will more likely ensure that public safety is better served without having to send large sections of the community for medical evaluation and disincentivising PA. All communities around the world may use the lessons from this study to consider validating their own such pre-participation tools. Of course, following such revision, there will be a need for significant efforts to implement and encourage use of the tool, even with digital and social media technology. These should allow and facilitate follow-up to assess the usefulness of such tools and empirically verify and validate their use for a safer sports environment.”
Comment 8: Ethics and References
- References are selected from current, relevant, and high-impact journals. Tables and figures are of sufficient quality and are self-explanatory. Ethical approval and conflict of interest declarations have been made as appropriate. Ethical approval has been obtained and stated. Informed consent processes have been explained. A conflict-of-interest declaration has been provided. Author contributions have been stated in detail.
Response 8a: Thank you.
- No data sharing policy has been stated.
Response 8b: Thank you for bringing this to our attention. This statement has been added to the manuscript just before the Reference section and states as follows:
Data Availability Statement: The raw data which forms the basis for the Tables in this manuscript is available on request to the corresponding author.
- Whether the study protocol has been previously registered (e.g. ClinicalTrials.gov) should be stated.
Response 8c. Thank you for this query. This study was not registered in Clinical Trials.gov because it was not a clinical trial. We had enquired about this prior to the conduct of the study but were advised that such registration was not necessary.
- More recent literature review can be done for some critical areas.
Response 8d. Thank you for the suggestion. References 12 and 23 has been updated with a more recent reference and a new reference 36 has been added. The later references have been re-numbered increasing the number of references to 39.
- The readability test results of the SPARQ questionnaire can be clearly stated in Figure 1.
Response 8e: Thank you for the suggestion. We have correctly retained the original Figure 1 as it appeared at the time the survey was conducted. However, the readability test scores for Figure 1 have been incorporated into the text: “The readability test scores for this questionnaire were Flesch-Kincaid Readability Grade Level 9 and Flesch Readability Ease score 45.2”.
- The statistical software version is stated but specific analysis packages are not. A data availability statement can be added.
Response 8f: We have replaced the words for the SPSS version with the official citation for use of IBM SPSS Statistics 21. The statistical analysis packages used are given in the third paragraph of the Data analysis section and we have updated this with mention of the reliability analysis determination. We have also added a data availability statement as mentioned earlier.
Round 2
Reviewer 3 Report
Comments and Suggestions for Authors
This article is an important contribution demonstrating the importance of public participation in the development of pre-physical activity screening tools. The authors' revisions, based on reviewer comments, have significantly improved the quality of the study.